# *Pseudomonas aeruginosa* lectin LecB impairs keratinocyte fitness by abrogating growth factor signalling

Alessia Landi[1,2] , Muriel Mari[3], Svenja Kleiser[1,2,4], Tobias Wolf[5], Christine Gretzmeier[4], Isabel Wilhelm[1,2,6], Dimitra Kiritsi[4] , Roland Thünauer[1,2], Roger Geiger[5] , Alexander Nyström[4], Fulvio Reggiori[3], Julie Claudinon[1,2], Winfried Römer[1,2,6]

**Lectins are glycan-binding proteins with no catalytic activity and ubiquitously expressed in nature. Numerous bacteria use lectins to efficiently bind to epithelia, thus facilitating tissue colonisation. Wounded skin is one of the preferred niches for *Pseudomonas aeruginosa*, which has developed diverse strategies to impair tissue repair processes and promote infection. Here, we analyse the effect of the *P. aeruginosa* fucose-binding lectin LecB on human keratinocytes and demonstrate that it triggers events in the host, upon binding to fucosylated residues on cell membrane receptors, which extend beyond its role as an adhesion molecule. We found that LecB associates with insulin-like growth factor-1 receptor and dampens its signalling, leading to the arrest of cell cycle. In addition, we describe a novel LecB-triggered mechanism to down-regulate host cell receptors by showing that LecB leads to insulin-like growth factor-1 receptor internalisation and subsequent missorting towards intracellular endosomal compartments, without receptor activation. Overall, these data highlight that LecB is a multitask virulence factor that, through subversion of several host pathways, has a profound impact on keratinocyte proliferation and survival.**

## Introduction

Bacteria can use many different strategies to infect host cells. In all cases, the initiation of an infection requires the recognition of specific structures at the host cell plasma membrane. This is often achieved by lectins, which bind to glycosylated residues on proteins and/or lipids present on the cell surface, mediating the attachment of the bacterium to the cell. Multivalency is an important feature of most lectins. On one hand, multivalency increases the binding affinity and specificity of the lectin–glycan interaction (Dam & Brewer, 2010). On the other hand, the binding of lectins to multiple cell surface receptors can induce receptor clustering and plasma membrane rearrangements, triggering their entry into the host (Römer et al, 2007; Windschiegl et al, 2009; Pezeshkian et al, 2017).

*Pseudomonas aeruginosa* is a Gram-negative bacterium, ubiquitously spread in nature. It is an opportunistic pathogen that can cause severe infections, especially in immunocompromised individuals, because of its resistance to most of the available antibiotics and its ability to form impenetrable biofilms. Hence, it has been classified in the "priority 1/critical" category of the World Health Organisation global priority pathogens list (global PPL) of antibiotic-resistant bacteria to promote the research and development of new antibiotic treatments (World Health Organization, 2017).

It is frequently implicated in hospital-acquired infections, where it has been reported to cause different types of infections. Wounded skin, after traumatic injuries, surgery or burns, is one of the preferentially targeted tissue by this bacterium, which has also been associated with the delay and prevention of wound healing. The presence of *P. aeruginosa* correlates in fact with a bad prognosis of healing, and leads to the persistence of the inflammatory stage of the wound healing process (Gjødsbøl et al, 2006; Bjarnsholt et al, 2007).

*P. aeruginosa* possesses two tetravalent lectins in its arsenal of virulence factors, LecA and LecB (also called PA-IL and PA-IIL, respectively). LecB is a tetramer, consisting of four monomers with high specificity for L-fucose and its derivatives (Garber et al, 1987; Gilboa-Garber et al, 2000). LecB production is regulated by *rhl* and *Pseudomonas* quinolone signal, which are part of the quorum-sensing systems (Winzer et al, 2000; Diggle et al, 2003). Once synthesised, LecB is exposed on the outer bacterial membrane, where it has been described to interact with the outer membrane porin OprF (Tielker et al, 2005; Funken et al, 2012).

[1]Faculty of Biology, Albert-Ludwigs-University Freiburg, Freiburg, Germany [2]Signalling Research Centres, Centre for Biological Signalling Studies and Centre for Integrative Biological Signalling Studies , Albert-Ludwigs-University Freiburg, Freiburg, Germany [3]Department of Biomedical Sciences of Cells & Systems, University of Groningen, University Medical Centre Groningen, Groningen, Netherlands [4]Department of Dermatology, Medical Center–University of Freiburg, Faculty of Medicine, Freiburg, Germany [5]Institute for Research in Biomedicine, Università della Svizzera Italiana, Bellinzona, Switzerland [6]Spemann Graduate School of Biology and Medicine, University of Freiburg, Freiburg, Germany

Correspondence: winfried.roemer@bioss.uni-freiburg.de

The current assumption is that LecB mainly functions by promoting the adhesion of *P. aeruginosa* to both the host cell and the exopolysaccharide matrix, which encases bacterial cells together. However, several in vitro and in vivo studies have shown LecB to act not only as an adhesin but also as an important virulence factor, capable of triggering additional host cell responses (Schneider et al, 2015; Wilhelm et al, 2019). LecB has been reported to be a determinant of *P. aeruginosa* cytotoxicity in lung epithelial cells and to block ciliary beating in human airways (Adam et al, 1997; Chemani et al, 2009). LecB-negative mutant bacteria exhibit an impaired biofilm formation in comparison with wild-type strains and no type VI pili assembly (Tielker et al, 2005; Sonawane et al, 2006). Furthermore, LecB induces alveolar capillary barrier injury in vivo, leading to a higher bacterial dissemination into the bloodstream (Chemani et al, 2009). Previous studies have reported additional effects of LecB in inhibiting cell migration and proliferation (Cott et al, 2016). However, its precise mechanism of action has not yet been elucidated and none of the existing studies have addressed its role in skin infections.

Here, we report that the *P. aeruginosa* lectin LecB is present in chronically infected human wounds, implying its contribution to the persistence of wound infections. Moreover, we show that insulin-like growth factor-1 receptor (IGF-1R) coprecipitates with LecB and that LecB leads to IGF-1R internalisation and missorting towards intracellular LC3-positive compartments. Notably, IGF-1R is internalised without being activated. We further demonstrate that LecB blocks the cell cycle and induces cell death, which is preceded by a strong vacuolisation. These vacuoles, which possess peculiar morphological features, originate from ruffle-like structures at subdomains of the plasma membrane where LecB is enriched. Therefore, we propose that LecB, in addition to play a role as an adhesion factor, both misregulates growth factor receptor signalling and subverts the endocytic system, leading to an impairment of vital keratinocyte functions.

## Results

### LecB is present in chronically infected human wounds

Although LecB is enriched in biofilms, which often characterise chronic wounds, no evidence had yet been provided showing its presence in wounds. Before we addressed the effects of LecB on human keratinocytes at a molecular level, we first verified the presence of LecB in infected human wounds. To this aim, we collected chronically wounded tissue from patients infected with *P. aeruginosa*, as shown by wound swabs. We stained the paraffin-embedded tissue sections with an antibody against *P. aeruginosa* to confirm its presence in the wounds. As control, we used normal skin samples. Indeed, we could detect the presence of *P. aeruginosa*, either as biofilm or in the form of small colonies (Fig 1A). We subsequently specifically stained for LecB, and strikingly, we found that this lectin was distributed in the wound sections, both in the keratinocyte layers and in the dermis (Fig 1B). In contrast, no LecB was detected in the normal skin control samples. This result provides the first evidence of LecB in chronic human wounds, with superinfection possibly playing a role in the wound chronicity.

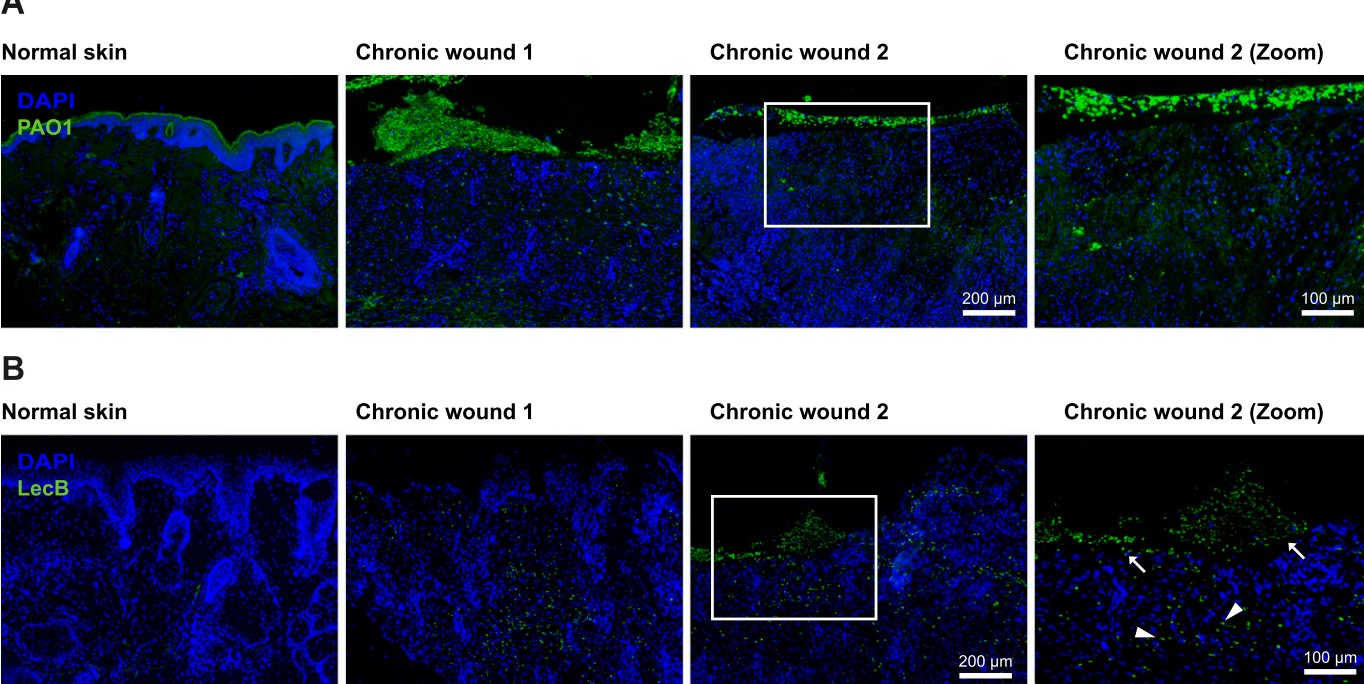

**Figure 1. LecB localisation in chronically infected human wounds.**
**(A, B)** Tissue sections of human infected wounds embedded in paraffin and stained for (A) *P. aeruginosa* (green) and for (B) LecB (green). Normal skin is used as negative control. Note: green signal in the upper left panel (A) is due to unspecific staining in the *stratum corneum*, not present in wounds. Rectangular squares refer to the zoomed area. Arrows point at LecB localised in the epidermal layers; arrowheads indicate LecB distributed in the dermis.

## LecB coprecipitates with essential plasma membrane receptors in normal human keratinocytes (NHKs)

Next, we moved to NHKs, the predominant cell type in the epidermis, to study the mechanism of action of LecB in molecular detail. LecB plays a crucial role in the adhesion of *P. aeruginosa* to host cells (Chemani et al, 2009), implying the necessity for the lectin to target plasma membrane receptors via their glycosylated residues. Therefore, we screened for potential LecB-interacting proteins via a pull-down assay. Briefly, we cultured NHKs in the presence or absence of 5 µg/ml (106 nM) of biotinylated LecB, lysed them under conditions that preserve protein–protein interactions and incubated with streptavidin agarose beads to precipitate LecB in complex with interacting proteins. Subsequently, we performed on bead digestion and analysed the obtained peptides by protein mass spectrometry. This analysis revealed the presence of important cell growth factor receptors within the enriched proteins in the LecB pull-down fractions (Table S1). The protein mass spectrometry results were verified by Western blot and confirmed the presence of two of the major keratinocyte growth factor receptors implicated in epidermal keratinocytes proliferation and migration, IGF-1R, and the epidermal growth factor receptor 1 (EGFR) (Haase, 2003; Sadagurski et al, 2005) (Figs 2A, and S1A and B). In addition to stimulating the cells with biotinylated LecB, an excess of L-fucose (30 mM) was added to clarify whether the interaction of LecB with IGF-1R or EGFR was carbohydrate dependent. As L-fucose saturates the carbohydrate-binding pockets of LecB, no binding to cell membranes could be detected when LecB and L-fucose were simultaneously applied to the cells, as neither IGF-1R nor EGFR could be coprecipitated (Fig S1A and B). We further investigated in detail the effect of LecB on IGF-1R because of its higher fold enrichment. We hypothesised that LecB interaction with IGF-1R triggers receptor internalisation. Indeed, surface staining experiments showed that IGF-1R is depleted from the plasma membrane of keratinocytes in a time-dependent manner as a consequence of LecB incubation (Fig 2B and C). Concurrently, we observed an intracellular redistribution of the receptor, which accumulates in regions positive for the lysosomal marker protein LAMP1 (Fig S1C). This observation correlated with the strong increase of lysosomes upon treatment (Fig S1D) and suggests that the receptor is targeted for degradation upon LecB. However, lysosomal degradation inhibition with bafilomycin A1 only partially restored IGF-1R levels (Fig S1E and F), indicating that LecB can act on other degradative pathways.

IGF-1R internalisation from the plasma membrane was also detectable after stimulation with 100 ng/ml of IGF-1 (Fig S2A and B). However, whereas IGF-1 induced a fast autophosphorylation of the receptor on multiple tyrosine residues (Tyr1135, Tyr1131, and Tyr1136), LecB did not (Fig 2D–G). Thus, we conclude that LecB does not lead to the activation of IGF-1R but rather triggers a "silent" receptor internalisation.

## LecB impairs cell survival signalling pathways and leads to cell cycle arrest

Growth factor receptors activate two major downstream signalling pathways, the mitogen-activated protein kinase/extracellular signal-regulated kinase (MAPK/ERK) and the phosphatidylinositol 3-kinase-protein kinase B (PI3K-AKT) cascade, which are responsible for inhibition of apoptosis, and stimulation of both protein synthesis and cell proliferation (Lemmon & Schlessinger, 2010; Mendoza et al, 2011). Whereas the former ultimately results in the phosphorylation and activation of extracellular signal-regulated kinase 1/2 (ERK1/2), the latter activates the mammalian target of rapamycin (mTOR). To determine whether these two signalling cascades were affected by LecB, we monitored ERK1/2 and mTOR phosphorylation at Thr202/Tyr204 and Ser2448, respectively. The amounts of phosphorylated proteins were normalised to the respective pan-proteins. Indeed, LecB did not lead to the phosphorylation of both ERK1/2 and mTOR (Fig 3A–D). ERK1/2 phosphorylation actually significantly decreased 1 h post-stimulation (Fig 3A and C). In contrast, LecB activated 5′ adenosine monophosphate–activated protein kinase (AMPK), a cellular energy and nutrient status sensor, which is activated in response to cellular energy depletion (Fig 3A and E). AMPK phosphorylation significantly increased 3 h post-stimulation and was inhibited by addition of L-fucose, confirming that this posttranslational modification is LecB dependent (Fig S3A and B).

The activation of AMPK correlates with the inhibition of ERK1/2 because the latter has been reported to negatively regulate the phosphorylation of AMPK via the liver kinase B1 (LKB1) (Woods et al, 2003; Shaw et al, 2004). Furthermore, consistently with the inhibition of ERK1/2 activity, we observed a strong cyclin D1 degradation (Fig 3F and G), which led to the arrest of cell cycle and, thus, to the reduction of cell viability (Fig 3H). Interestingly, the cytotoxic effect was preceded by an extensive cytoplasmic vacuolisation (Fig 3I). L-fucose supplementation rescued the phenotype and restored cell viability (Fig 3H and I). Altogether, these observations elucidate that LecB impairs a key mechanism that regulates cell survival and proliferation, resulting in the blockage of cell cycle.

## The cytotoxic effect of LecB is accompanied by the formation of LecB-positive intraluminal vesicle-containing vacuoles

Next, we used transmission electron microscopy to further investigate the cellular vacuolisation triggered by LecB treatment. Untreated keratinocytes presented several light content vacuoles (Category 1), corresponding to endo-lysosomal compartments (Fig 4A). In contrast, LecB-treated cells displayed numerous degradative vesicles (Category 2) and an additional type of vacuoles (Category 3), with irregular shapes and variable sizes (Fig 4B). Surprisingly, these vacuoles contained a large number of clearly defined intraluminal vesicles (Fig 4B) and they appeared to cluster together to form an intricate network (Fig 4B, zoom). To get insights into the origin of these structures, we performed a time-course experiment by collecting cells at different time points, from 30 min to 12 h after exposure to LecB (Fig S4A–E′). The Category 3 vacuoles were detectable after 30 min of LecB incubation, but they were quite rare. However, they became more prominent 3 h post-LecB exposure (Fig S4D). Interestingly, specific regions of the plasma membrane of the LecB-treated keratinocytes were extremely irregular and characterised by the presence of ruffle-like structures (Fig S4A–E). As the number of Category 3 vesicles located at these regions increased over time, we hypothesised that the intraluminal vesicle-containing vacuoles originate from subdomains of the plasma membrane upon LecB interaction with surface proteins. This hypothesis implies that LecB is

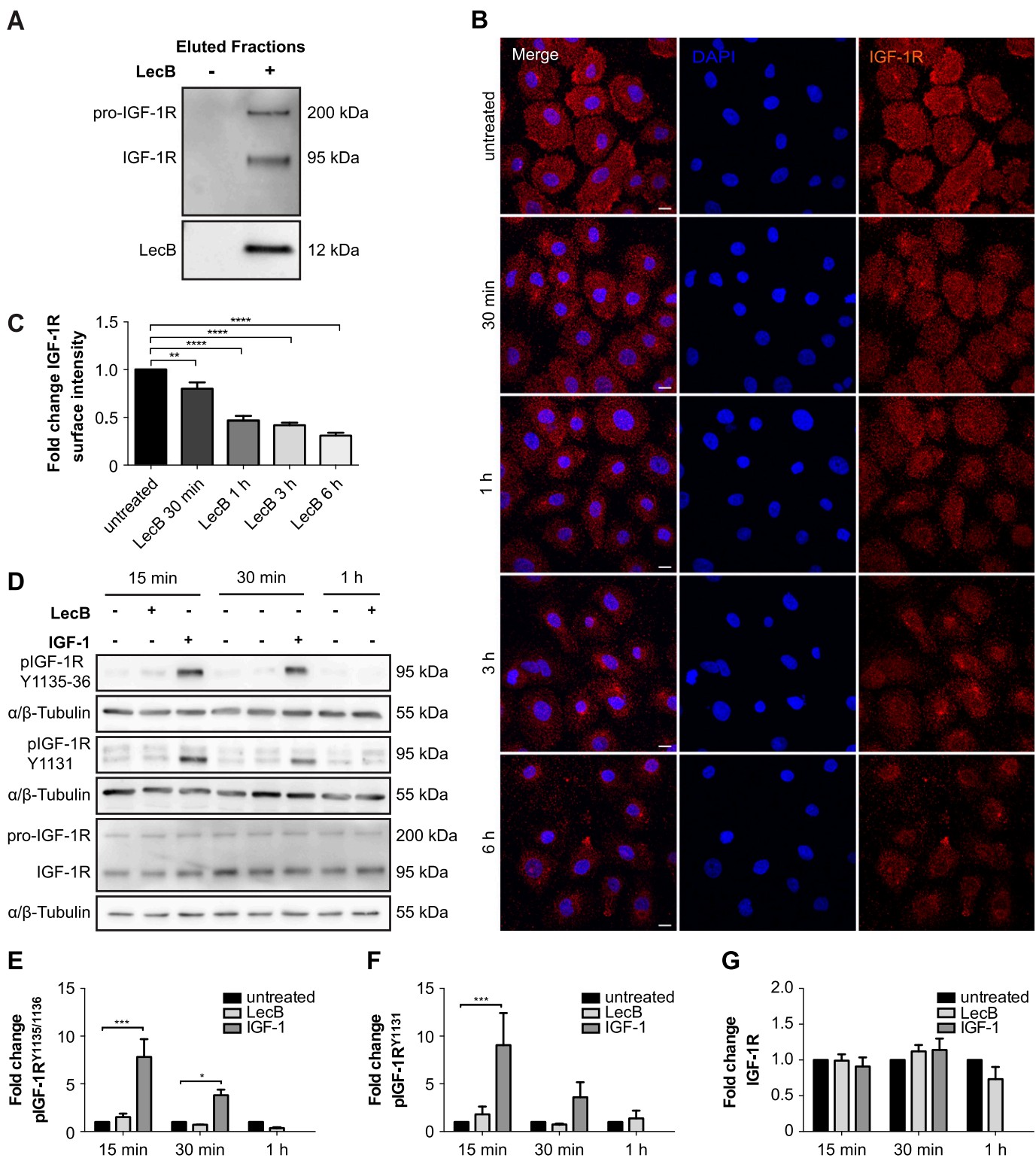

**Figure 2. LecB depletes IGF-1R from the plasma membrane without inducing its activation.**
**(A)** Western blot of eluted samples from pull-down assay. NHKs were stimulated in the presence of 5 $\mu$g/ml biotinylated LecB (106 nM). The lysates were incubated with streptavidin beads and further eluted. Western blot was performed and IGF-1R and LecB were detected in the precipitated fractions. **(B, C)** Surface staining of keratinocytes treated with LecB (5 $\mu$g/ml) for the indicated time points. **(B)** Images show maximum intensity projections. After stimulation, the cells were stained for IGF-1R (red) and DAPI (blue). Scale bar: 10 $\mu$m. **(C)** Quantification of IGF-1R surface intensity from n = 4 independent experiments. Bars display the mean value ± SEM. ** denotes $P < 0.01$; **** denotes $P < 0.0001$; one-way ANOVA was used for statistical analysis. **(D)** Representative blots from lysates after LecB (5 $\mu$g/ml) and IGF-1 (100 ng/ml) stimulation for the indicated times. Antibodies against different phosphorylation sites of IGF-1R (Tyr1131 and Tyr1135/1136)

present in the Category 3 vacuoles. To test this, we exposed keratinocytes to biotinylated LecB before performing immunoelectron microscopy (IEM) using anti-biotin antibodies (Fig 4C–F). Notably, we specifically detected LecB on both the limiting membrane of the Category 3 vacuoles and to a minor extent on their intraluminal vesicles (Fig 4D). Moreover, we also identified biotinylated LecB at the plasma membrane, where it was especially enriched in the ruffle-like regions (Fig 4F), supporting the notion that LecB drives the formation of a plethora of intracellular vesicles originating from the plasma membrane.

### LecB is trafficked towards an endocytic degradative route

Next, we proceeded to characterise the intracellular LecB-positive vacuoles detected by IEM in immunofluorescence experiments, where keratinocytes exposed to fluorescently labelled LecB were stained with antibodies recognising different organelle marker proteins. We found a time-dependent colocalisation of LecB with microtubule-associated proteins 1A/B light chain 3B (LC3) and ras-related protein Rab-9A (RAB9A), which are generally enriched on autophagosomes and late endosomes, respectively (Fig 5A and B). Moreover, but to a minor extent, we also observed a time-dependent colocalisation between LecB and lysosomal-associated membrane glycoprotein 1 (LAMP1) (Fig S5A). In contrast, no colocalisation with the recycling endosome marker protein ras-related protein Rab-11 (RAB11) was detected (Fig S5B). Endocytosis and internalisation of LecB was also assessed biochemically by analysing the ubiquitination of the proteins associated with LecB, as a large number of growth factor receptors undergo ubiquitination upon endocytosis (Sehat et al, 2008; Haglund & Dikic, 2012). In a time-course pull-down experiment, we detected a decrease of the coprecipitated levels of IGF-1R and an increase of the ubiquitin levels in the eluted fraction, indicating that LecB-interacting partners are ubiquitinated, thus very likely internalised (Fig 5C and D).

Taken together, these results demonstrate that LecB is trafficked towards degradative compartments and suggest that ubiquitin may be the endocytic signal for the degradation of IGF-1R and potentially other LecB-interacting partners.

### LAP participates in targeting LecB–IGF–1R complexes to degradation

As we saw a time-dependent increase in colocalisation between LC3 and LecB structures (Fig 5A), we decided to investigate this further. LC3 is an ubiquitin-like protein that upon autophagy induction is converted from a cytoplasmic form (i.e., LC3-I) to one associated with the autophagosomal membranes (i.e., LC3-II) through conjugation to phosphatidylethanolamine (Kabeya et al, 2004). LC3-II can also be generated on the limiting membrane of endosomes during processes such as LC3-associated phagocytosis (LAP) (Sanjuan et al, 2007; Florey et al, 2011). The fact that LecB was detected on single-membrane vacuoles (autophagosomes are double-membrane vesicles with cytoplasmic content) that are

positive for RAB9 (Figs 4D and 5B), inferred in an activation of LAP-like pathway. Although induction of both autophagy and LAP induces the formation of LC3-II, only autophagy leads to the degradation of this conjugate (Tanida et al, 2005). Indeed, we observed a time-dependent increase of LC3-II levels upon exposure of keratinocytes to LecB (Fig 6A and B). However, this protein was not turned over because no difference in LC3-II levels could be observed when LecB-treated keratinocytes were incubated with either bafilomycin A1 or cycloheximide to assess LC3-II stability (Fig S6A–D). To clarify the origin of LC3-II, we silenced ATG13 (Fig S6E and F), an essential component of the ULK kinase complex, which is required for macroautophagy (Ganley et al, 2009) and dispensable for LAP (Florey et al, 2011; Martinez et al, 2011). Our results revealed that LC3-II formation upon cell treatment with LecB is ATG13 independent (Fig S6E and G). Furthermore, when we performed LecB treatment in combination with cytochalasin D, an inhibitor of actin polymerization, we found a decrease in LecB uptake (Fig S7A and B), indicating that actin is important for LecB internalisation.

Next, we investigated whether IGF-1R also colocalised with LC3-positive vesicles. Indeed, we found colocalisation between IGF-1R, LC3, and LecB (Fig 6C), but interestingly, we did not observe LC3 recruitment onto membranes when cells were treated exclusively with IGF-1 (Fig S8). Finally, we examined another receptor, the transferrin receptor (TfR), which was not pulled down by LecB (Fig S9). Unlike IGF-1R and other growth factor receptors, which can be internalised via clathrin-independent mechanisms (Boucrot et al, 2015), TfR enters the cells via clathrin-mediated endocytosis and it is recycled back to the plasma membrane upon iron delivery (Hopkins & Trowbridge, 1983). Interestingly, whereas we observed colocalisation between IGF-1R and LC3 (Fig 6D), we did not detect changes in TfR distribution nor LC3 colocalisation upon LecB treatment (Fig 6E). This is supported by the absence of colocalisation between LecB and RAB11 compartments as well (Fig S5B) and suggests that LecB preferentially commits receptors to degradation, thereby affecting their signalling.

Taken together, these results demonstrate that LC3 recruitment onto LecB–IGF–1R complex–positive endosomes is independent from macroauthophagy.

## Discussion

Bacterial lectins have predominantly been described as adhesion proteins, which promote the attachment of bacteria to cell plasma membrane through their interaction with host cell surface glycans (Sharon, 1987). This study provides new insights into the role of *P. aeruginosa* lectin LecB as virulence factor that impairs growth factor receptor signalling and trafficking, thereby affecting keratinocyte fitness. Pull-down studies showed the co-isolation of LecB with several plasma membrane receptors in keratinocytes, among which IGF-1R and EGFR were two of the principal interactors. We further demonstrated that LecB depletes IGF-1R from the plasma

---

and against total IGF-1R were used. Tubulin was used as loading control. **(E, F, G)** Blot quantification. Phosphorylated IGF-1R (E, F) and total IGF-1R levels (G) are represented as fold change compared with the loading control from n = 3 independent experiments. * denotes $P < 0.05$; *** denotes $P < 0.001$; two-way ANOVA was used for statistical analysis.

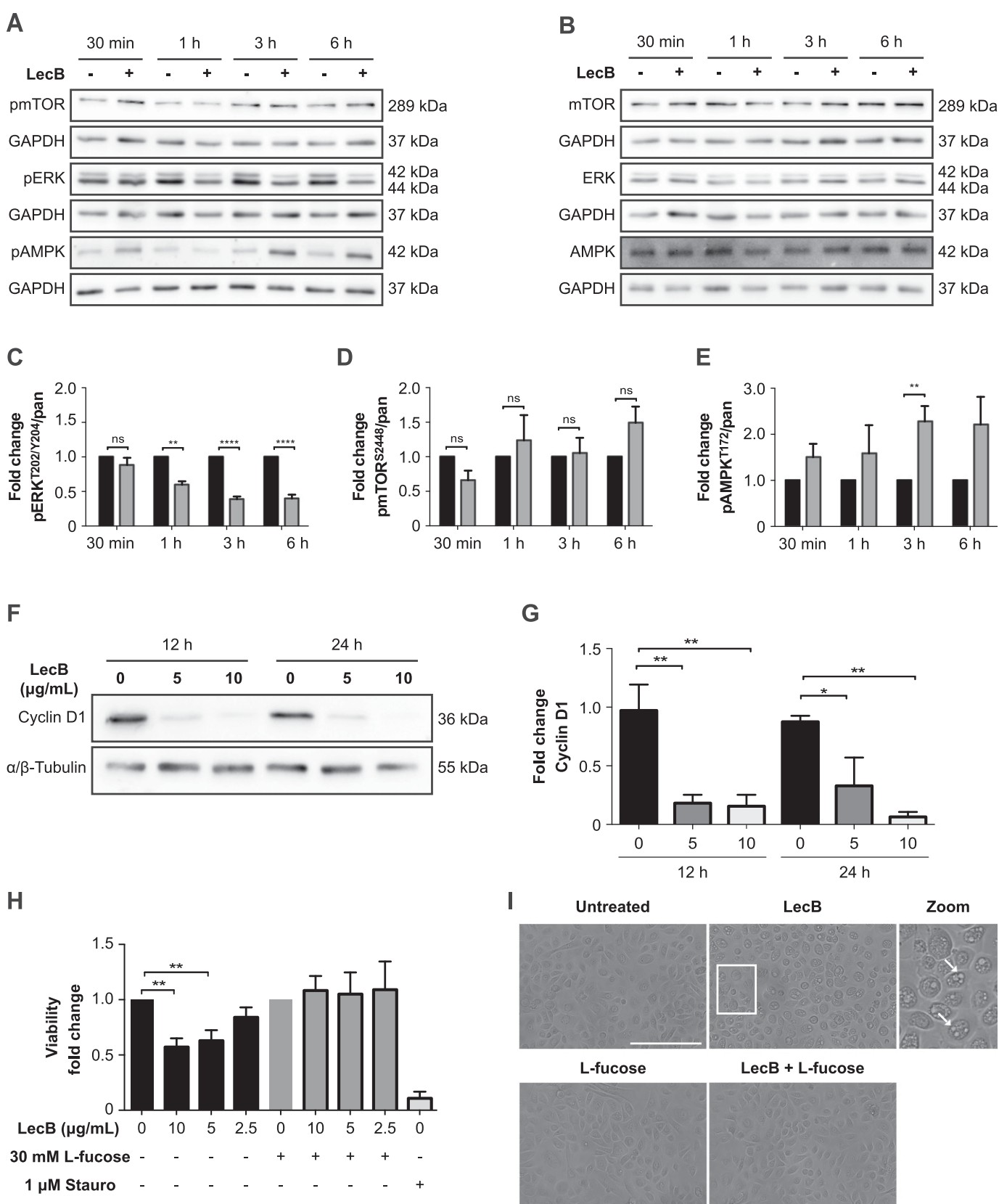

membrane by inducing its internalisation. To modulate signalling, IGF-1R (as well as EGFR) needs to be activated by ligand binding and consequent autophosphorylation on three tyrosine residues (i.e., Tyr1135, Tyr1131, and Tyr1136) in its C-terminal tail (Favelyukis et al, 2001; Sousa et al, 2012). Interestingly, unlike IGF-1, LecB triggers IGF-1R endocytic trafficking towards degradative compartments without receptor activation.

PI3K/AKT and MAPK/ERK signalling cascades have been intensively investigated for their role in promoting cell and organismal growth upon growth factor signals. This is mainly achieved by AKT-mediated activation of mTOR, which is a key regulator of anabolic processes necessary during cell growth (Nave et al, 1999; Dennis et al, 2001) and by the phosphorylation of ERK1/2, which results in cell proliferation (Rossomando et al, 1991; Meloche & Pouysségur, 2007). We found that neither mTOR nor ERK1/2 is activated upon cell exposure to LecB. Specifically, we observed a decrease in the levels of phospho-ERK1/2 after 1 h of LecB treatment, which correlates with an increase of phospho-AMPK. Reciprocal feedback loops have been described to operate between the AMPK and the MAPK/ERK pathways and, whereas ERK1/2 activation was reported to be attenuated upon stimulation of WT MEFs with the AMPK activator A-769662, little effect on ERK1/2 phosphorylation was detected when AMPK-null MEFs were used (Zheng et al, 2009; Shen et al, 2013). The suppression of cell survival signalling pathways can also clarify the arrest of the cell cycle in the G1/S transition observed upon LecB incubation. ERK1/2 activity is in fact required for both the cell cycle entry and the suppression of negative cell cycle regulators. Specifically, cyclin D1 expression is enhanced by the activation of the MAPK/ERK pathway (Lavoie et al, 1996; Weber et al, 1997), via the transcription factors c-Fos (Burch et al, 2004) and Myc (Seth et al, 1991). Cells, entering into the S phase show low levels of cyclin D1, which is again induced and reaches high levels to ensure the progression into the G2 phase (Yang et al, 2006). LecB stimulation mediates a strong reduction of cyclin D1, whose levels are not restored over time, resulting in the arrest of the cell cycle and subsequential induction of cell death. Therefore, we speculate that LecB, despite devoid of catalytic activity, may contribute in promoting tissue damage to facilitate bacterial dissemination and extracellular multiplication during wound infections. Furthermore, it is possible that LecB induces a similar fate to other cell types, including immune cells, thus further promoting the infection. Future work will provide new insights on the impact of LecB on the immune response during chronic wound infections.

Interestingly, the cytotoxic effect of LecB was preceded by an extensive cytoplasmic vacuolation. Electron microscopy inspection of LecB-treated cells revealed the peculiar nature of these vacuoles that accumulate over time and display an increased number of intraluminal vesicles. We also show that these vacuoles probably originate from ruffle-like plasma membrane regions where LecB was also found to be enriched. In the case of other bacterial lectins, such as the B-subunit of Shiga toxin (StxB) from *Shigella dysenteriae* and LecA from *P. aeruginosa*, the sole interaction with the glycosphingolipid receptor globotriaosyl ceramide (Gb3) is sufficient to drive membrane shape changes leading to their subsequent uptake (Römer et al, 2007; Eierhoff et al, 2014). In the case of LecB, however, pull-down experiments point to the existence of several receptors, also attributable to the fact that fucosylation is a very common modification. LecB exists as a tetramer and each monomer possesses a binding pocket with the highest affinity to L-fucose and its derivatives (Garber et al, 1987; Sabin et al, 2006). Although the L-fucose dissociation constant is in the micromolar range (2.9 $\mu$M), an increase in avidity can be achieved by a higher degree of interactions. This implies that LecB might crosslink several different surface receptors, thus inducing a higher degree of membrane rearrangements that could explain the extensive alterations at the plasma membrane. Moreover, multiple interactions could provide the bacterium with additional resources for the initiation of host tissue colonisation.

By following LecB trafficking, we sought to characterise the nature of LecB-mediated vacuoles, which, from a structural point of view, shows similarities with multivesicular bodies, as they contain numerous intraluminal vesicles. Immunofluorescence experiments revealed a time-dependent increase of colocalisation between LecB and Rab-9, LC3, and LAMP-1. Therefore, our data indicate that LC3 is recruited on LecB containing late endosomes, which may further favour their lysosomal degradation through a process that may be similar or identical to LAP. LAP can be triggered by several receptors. In addition to toll-like receptors, whose signalling during phagocytosis rapidly recruits LC3 to phagosomes, the phosphatidylserine receptor TIM4, or the C-type lectin dectin-1 can also induce LAP to efficiently clear dead cells and to facilitate antigen presentation, respectively (Sanjuan et al, 2007; Martinez et al, 2011; Ma et al, 2012). LC3 can associate with single-membrane phagosomes even in the absence of pathogens or dead cells, such in the case of phagocytosis and degradation of photoreceptor outer segments by retinal pigment epithelium or during the secretion of mucins in goblet cells (Kim et al, 2013; Patel et al, 2013). Our data indicate that IGF-1R colocalises with LecB and LC3, suggesting that LAP or an LAP-like pathway is involved in IGF-1R sorting for degradation, after LecB-induced receptor internalisation.

The current model describes that tyrosine kinase receptors, upon ligand binding, are internalised and trafficked to early endosomes.

**Figure 3.   LecB impairs cell survival signalling pathways and leads to cell cycle arrest.**
**(A, B, C, D, E)** Representative blots and relative quantifications of N = 3 independent experiments. NHKs were treated with 5 $\mu$g/ml LecB for the indicated time and lysates were subjected to SDS–PAGE and Western blot analysis using the designated anti-phospho (A) and anti-pan antibodies (B). GAPDH was added as loading control. Graphs (C, D, E) depict the fold change of the phosphorylated protein compared with pan levels and represent the mean value ± SEM. ** denotes $P < 0.01$, **** denotes $P < 0.0001$, ns denotes not significant; multiple $t$ tests were used for statistical analysis. **(F)** Western blot showing cyclin D1 levels after 12 and 24 h of LecB stimulation (5 or 10 $\mu$g/ml). **(G)** Quantification of cyclin D1 relative to tubulin. The mean value ± SEM of N = 3 independent experiments is reported. * denotes $P < 0.05$, ** denotes $P < 0.01$; one-way ANOVA was used for statistical analysis. **(H)** MTT assay assessing the cytotoxic effect of LecB. NHKs were treated with the indicated LecB concentration with or without 30 mM L-fucose for 24 h. Staurosporine (1 $\mu$M) was used as positive control. MTT was added to the medium and left for 4 h. The absorbance at 570 nm was measured and plotted as fold change compared with the untreated or L-fucose–treated sample. The mean value ± SEM of N = 5 is plotted. ** denotes $P < 0.01$; one-way ANOVA was used for statistical analysis. **(I)** Representative images of keratinocyte monolayers after 24-h exposure to 5 $\mu$g/ml LecB with or without 30 mM L-fucose. White arrows in the zoomed image point at vacuolar structures induced by lectin treatment. Scale bar: 200 $\mu$m.

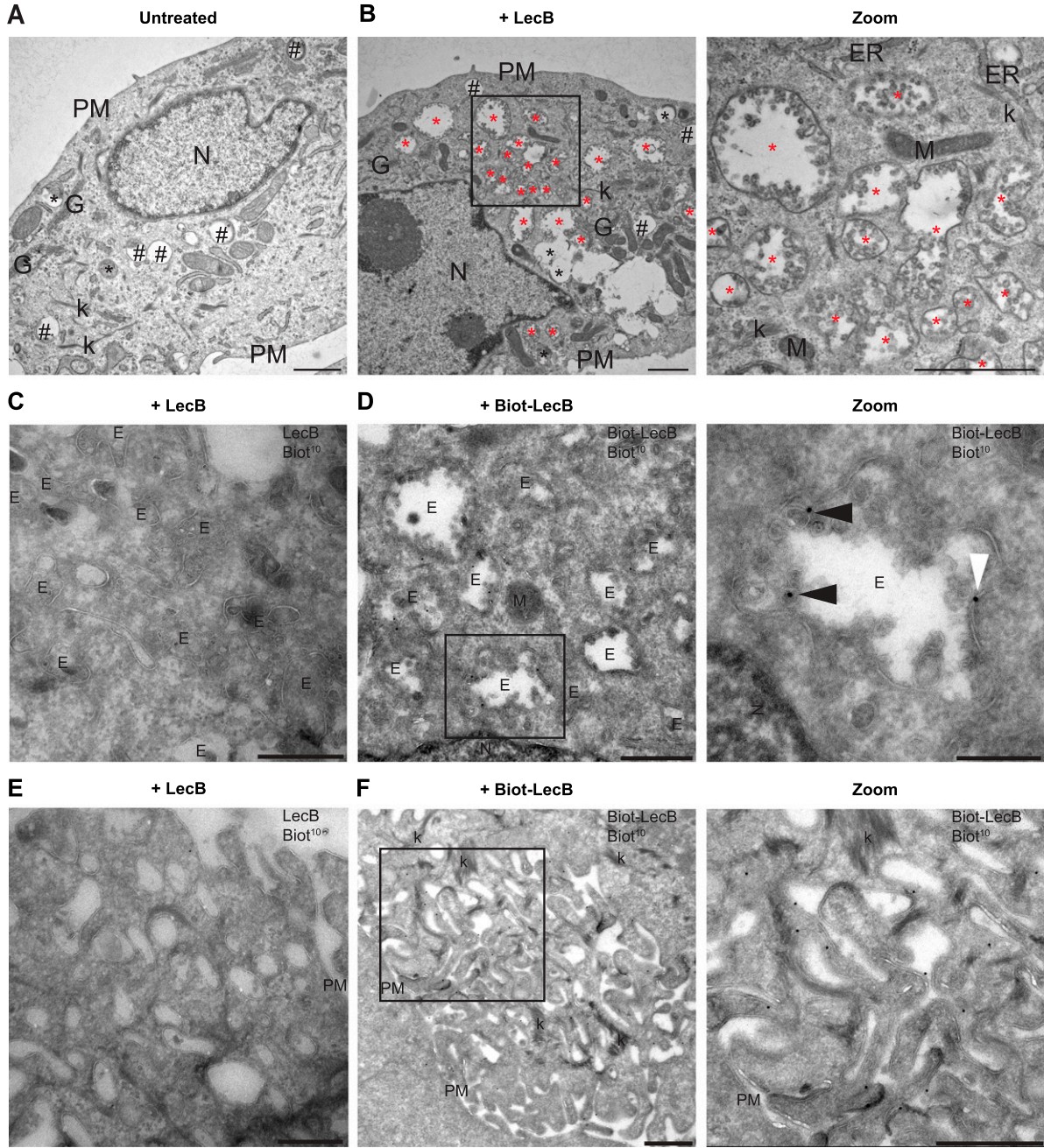

**Figure 4. The cytotoxic effect of LecB is preceded by the formation of intraluminal vesicle-containing vacuoles.**
**(A, B)** NHKs were incubated with 5 μg/ml LecB and processed for conventional electron microscopy at 12 h post-incubation. **(A, B)** Representative electrographs of the untreated (A) and LecB-treated (B) cells. Black hashtags point at Category 1 vacuoles, whereas black and red asterisks indicate Category 2 and Category 3 vacuoles, respectively. Zoomed image of panel (B) shows a higher magnification of the Category 3 vacuoles containing intraluminal vesicles. Scale bars: 1 μm. **(C, D, E, F)** Cells treated with 5 μg/ml biotinylated LecB or un-tagged LecB for 12 h were subjected to immuno-EM. **(C, E)** Control staining showing no aspecific antibody binding. **(D)** LecB localises in internal vesicles, either in the lumen (panel [D] zoom, black arrowheads) or at the limiting membrane (white arrowheads). **(F)** LecB localisation at the plasma membrane, in the ruffle-like region. Scale bars: 500 nm. Scale bar zoom panel (D): 200 nm. E, endosome; ER, endoplasmic reticulum; G, Golgi apparatus; k, keratin; M, mitochondrion; N, nucleus; PM, plasma membrane.

From here, they can either be sorted to lysosomal degradation via multivesicular bodies or they can be recycled back to the plasma membrane. The equilibrium between IGF-1R degradation and recycling is essential to modulate receptor signalling (Morrione et al, 1997; Monami et al, 2008). Our data demonstrate that IGF-1R trafficking is subverted upon cell treatment with LecB, which induces the sorting of this receptor towards degradative routes, without activating it. We do not know yet whether there is a direct or indirect interaction between LecB and IGF-1R, despite being co-isolated as complex. However, because IGF-1R is not the sole protein interacting with LecB in keratinocytes, we speculate that this "targeting for degradation" strategy can be valid for other plasma membrane receptors as well

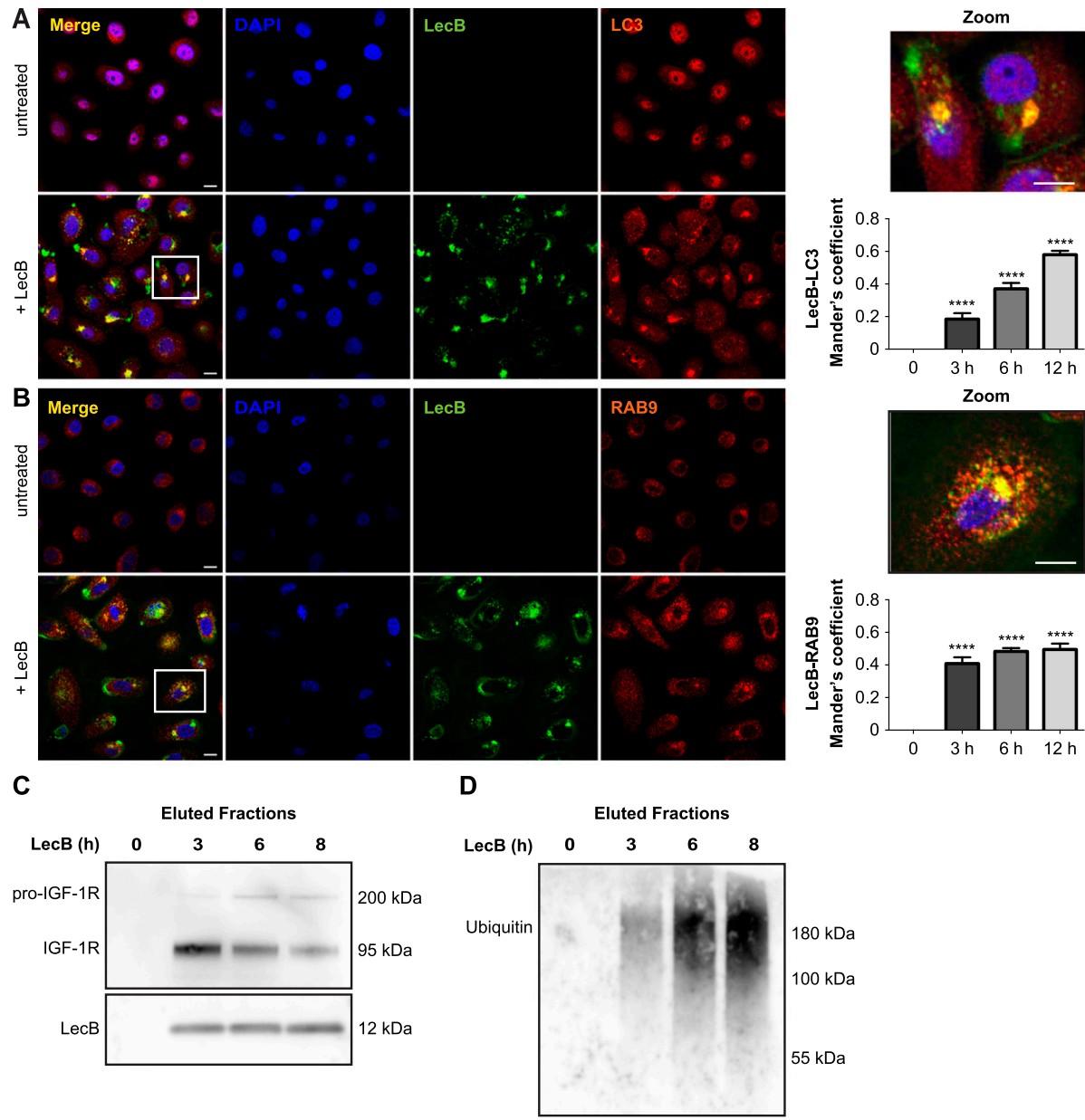

**Figure 5. LecB localises in LC3- and RAB9-positive compartments.**
**(A, B)** Confocal micrographs with respective colocalisation quantification of keratinocytes stimulated with 5 μg/ml of fluorescently labelled LecB (green). Panels indicated as "+LecB" refer to 12 h incubation. **(A, B)** After fixation and permeabilisation, the cells were stained for LC3 (A) and RAB9 (B) (both in red). Graphs report the mean value ± SEM of Mander's overlap coefficients calculated from at least three independent experiments. **** denotes *P* < 0.0001; one-way ANOVA was used for statistical analysis. Scale bar: 10 μm. **(C, D)** Western blot of eluted samples from time series pull-down assays using biotinylated LecB (5 μg/ml). **(C, D)** Membranes were probed for LecB, IGF-1R (C) and ubiquitin (D).

(e.g., EGFR) and that it can be used by *P. aeruginosa* to silence host cell receptors to favour tissue colonisation in wounds.

Wound infections represent a socioeconomic burden for the health care system and, given that *P. aeruginosa* is very hard to eliminate with the available antibiotics, there is urgent need for the development of alternative therapeutic strategies (Norberg et al, 2017; Sommer et al, 2018). Our findings shed new light on the *P. aeruginosa* lectin LecB, showing that it is capable of inducing a succession of cellular events, despite being devoid of catalytic activity and by virtue of its sole capability to bind to sugar moieties

on the plasma membrane. This knowledge paves the way for a better understanding of *P. aeruginosa* wound infections.

# Materials and Methods

### Antibodies, inhibitors, and activators

Used antibodies are listed in Tables S2 and S3. Aprotinin (0.8 μM), leupeptin (11 μM), and pefabloc (200 μM), used as protease inhibitors,

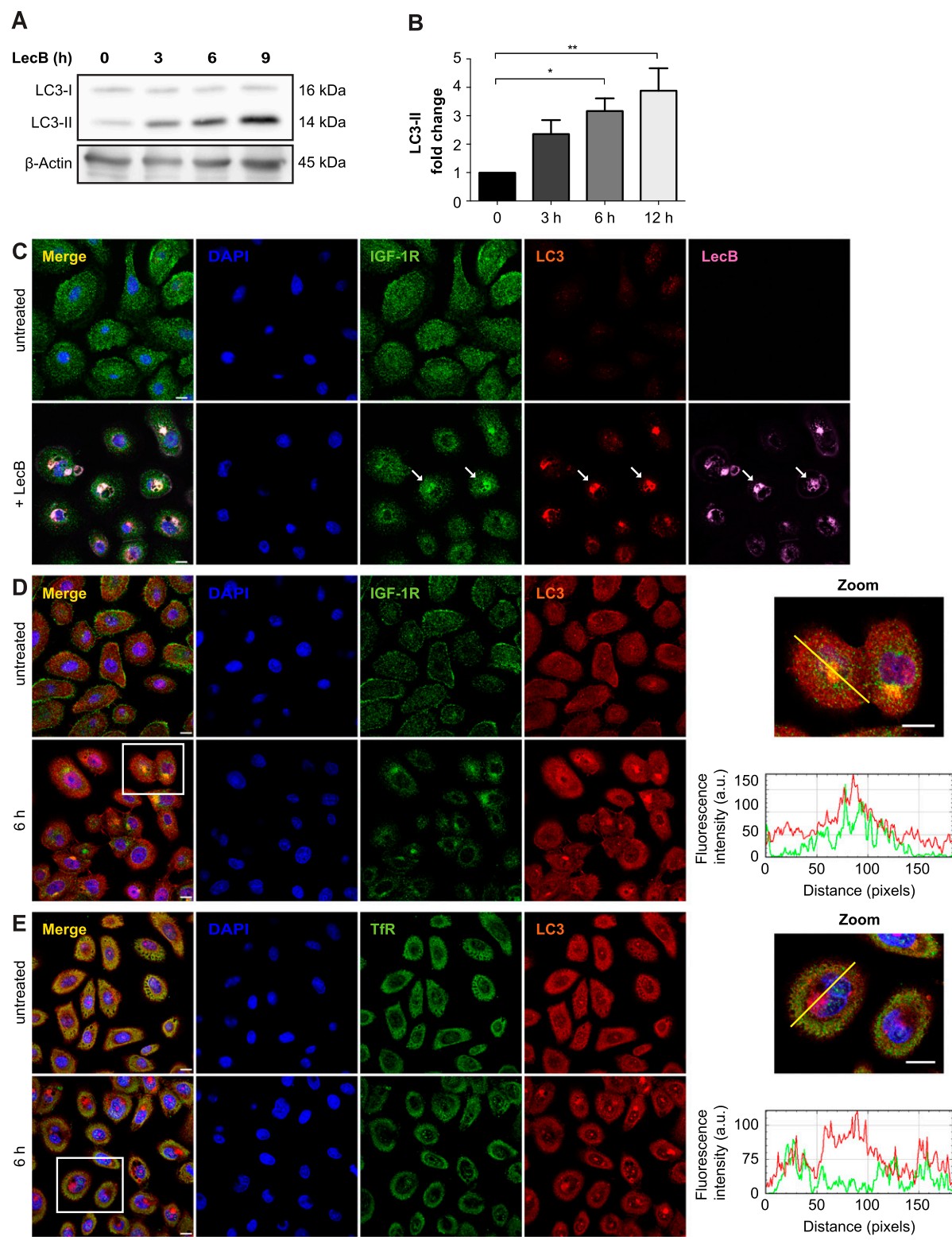

**Figure 6. IGF-1R is sorted to degradation by LecB.**
**(A, B)** NHKs were incubated with LecB (5 μg/ml) for the indicated time points and whole cell lysates were immunoblotted for LC3 and β-actin (A). **(B)** The levels of LC3-II were normalised to β-actin and depicted as fold change increase to the untreated condition (B). Error bars indicate means ± SEM of N = 5 independent experiments. * denotes $P < 0.05$, ** denotes $P < 0.01$; one-way ANOVA was used for statistical analysis. **(C)** Confocal micrographs of keratinocytes treated with 5 μg/ml of fluorescently labelled LecB (pink) and stained for LC3 (red) and IGF-1R (green). Panel indicated as "+LecB" refers to 12 h incubation. White arrows point at colocalisation among LecB, LC3, and IGF-1R. Scale bar: 10 μm. N = 3. **(D, E)** Representative images of NHKs treated with LecB and stained for (D) IGF-1R (green), LC3 (red), and (E) TfR (green). Intensity profiles of single cells, along the yellow line are shown. Scale bar: 10 μm.

and phosphatase inhibitor cocktail 3 (1:100) were from Sigma-Aldrich. To block LecB binding to the host cell membranes, L-fucose (Sigma-Aldrich) was used at a concentration of 30 mM. Cycloheximide (20 μg/ml) and staurosporine (500 nM) were both purchased from Sigma-Aldrich and used as a protein synthesis inhibitor and as positive control for cell death, respectively. Bafilomycin A1 was from InvivoGen and used at a concentration of 100 or 200 nM, whereas cytochalasin D was from Sigma-Aldrich and applied at a concentration of 0.5 μg/ml. Human IGF-1 was from Thermo Fisher Scientific and used as an IGF-1 receptor activator.

### Cell culture

NHKs, kindly provided by D Kiritsi (Department of Dermatology), were grown at 37°C and in the presence of 5% $CO_2$ in keratinocyte medium supplemented with bovine pituitary extract, epidermal growth factor, and 0.5% penicillin/streptomycin. Cells were seeded 24 h before each experiment. If not stated differently, all treatments were performed in complete medium.

### LecB production and in vitro labelling

Recombinant LecB (UniProt ID: Q9HYN5_PSEAE) was produced from *Escherichia coli* BL21(DE3) containing pET25pa21 and purified via chromatography using a mannose-agarose column as previously described (Mitchell et al, 2002; Chemani et al, 2009). The eluate was dialysed in water for 7 d and finally lyophilised. The obtained powder was dissolved in PBS (138 mM NaCl, 2.7 mM KCl, 8 mM $Na_2HPO_4$, and 1.5 mM $KH_2PO_4$, pH 7.4) and sterile-filtered. In addition, to exclude endotoxin contamination, LecB was further purified using an endotoxin removal column (Hyglos) and an LAL chromogenic endotoxin quantitation kit (Thermo Fisher Scientific) was carried out. The purity of the lectin was confirmed by SDS–PAGE. If not stated differently, LecB was used at a final concentration of 5 μg/ml (106 nM).

LecB was fluorescently labelled using Cy3 (GE Healthcare) or Alexa Fluor488 (Thermo Fisher Scientific) monoreactive NHS ester and purified using Zeba Spin desalting columns (Thermo Fisher Scientific) according to the manufacturers' instructions.

Biotinylated LecB was obtained using sulfo-NHS-SS-Biotin (Thermo Fisher Scientific) according to the supplier's instructions and dialysed two times for 1 h in water and one time overnight in PBS.

### Gene silencing

Cells were incubated with a mix of siRNA against ATG13 (Dharmacon) and transfection reagent (Dharmacon) for 5–6 h. 24 h post-transfection lysates were checked for silencing efficiency. LecB stimulation was performed 36 h post-silencing. A control siRNA (Santa Cruz) was used to evaluate off-target effects.

### Mass spectrometry analyses

Cells were seeded in T-75 flasks and stimulated with biotinylated LecB at a concentration of 5 μg/ml. Upon treatment, the cells were washed in PBS and lysed with 50 mM Tris–HCL, pH 7.5, 150 mM sodium chloride, 1% (vol/vol) IGEPAL CA-630, and 0.5% (wt/vol) sodium deoxycholate in water. The protein lysates were incubated overnight at 4°C with streptavidin agarose beads (Thermo Fisher Scientific) to precipitate LecB–biotin–protein complexes. The beads were washed four times in PBS and then resuspended in 8 M urea. Immunoprecipitates were predigested with LysC (1:50, w/w) followed by the reduction of disulfide bonds with 10 mM DTT and subsequent alkylation with 50 mM iodacetamide. Then samples were diluted 1:5 with ammonium bicarbonate buffer, pH 8, and trypsin (1:50, w/w) was added for overnight digestion at RT. The resulting peptide mixtures were acidified and loaded on C18 StageTips (Rappsilber et al, 2007). Peptides were eluted with 80% acetonitrile (ACN), dried using a SpeedVac centrifuge, and resuspended in 2% ACN, 0.1% TFA, and 0.5% acetic acid.

Peptides were separated on an EASY-nLC 1200 HPLC system (Thermo Fisher Scientific) coupled online to a Q Exactive HF mass spectrometer via a nanoelectrospray source (Thermo Fisher Scientific). Peptides were loaded in buffer A (0.5% formic acid) on in-house–packed columns (75-μm inner diameter, 50-cm length, and 1.9-μm C18 particles from Dr. Maisch GmbH). Peptides were eluted with a nonlinear 270-min gradient of 5–60% buffer B (80% ACN, 0.5% formic acid) at a flow rate of 250 nl/min and a column temperature of 50°C. The Q Exactive HF was operated in a data-dependent mode with a survey scan range of 300–1,750 m/z and a resolution of 60,000 at m/z 200. MaxQuant software (version 1.5.3.54) was used to analyse MS raw files (Cox & Mann, 2008). MS/MS spectra were searched against the human Proteome UniProt FASTA database and a common contaminants database (247 entries) by the Andromeda search engine (Cox et al, 2011).

### Immunofluorescence and confocal microscopy

Between 4 and 5 × $10^4$ cells were seeded on 12-mm glass cover slips in a 24-well plate and cultured for 1 d before the experiment. Keratinocytes were treated with LecB for the indicated concentrations and time points. Surface or whole cell staining was then performed. For surface staining, NHKs were fixed with 4% (wt/vol) formaldehyde (FA) for 10 min and quenched with 50 mM ammonium chloride for 5 min. The cells were blocked with 3% (wt/vol) of BSA in PBS and incubated overnight with the primary antibody diluted in blocking solution (see the details for the used antibodies in Table S2). The cells were then washed in PBS and incubated for 30 min with the dye-labelled secondary antibody. For whole cell staining, the cells were fixed with 4% (wt/vol) FA and quenched, as previously described. When methanol fixation was used instead of FA, the cells were incubated with ice cold methanol for 10 min and rinsed twice with PBS. After fixation, 10 min incubation with 0.1% (vol/vol) Triton X-100 in PBS was performed. The cells were blocked and subsequently stained with the primary and secondary antibody, as previously described, but using blocking solution supplemented with 0.05% Triton X-100.

Nuclei were additionally stained with DAPI diluted in PBS with 0.1% (vol/vol) Triton X-100. After three additional washes, glass cover slips were mounted with Mowiol and imaged using an A1R Nikon confocal microscopy system with a 60× oil immersion objective (NA = 1.49). Z-stacks of at least three different areas per condition were acquired and analysed with Fiji ImageJ 1.0 software. Coloc2 Fiji's plugin was used for colocalisation analysis.

### Western blot analyses

NHKs were seeded in a 12-well plate at a density of $1.3 \times 10^5$ per well and stimulated with LecB for the indicated time points. After the treatment, the cells were washed with PBS and lysed in RIPA buffer (20 mM Tris–HCl, pH 8, 0.1% [wt/vol] SDS, 10% [vol/vol] glycerol, 13.7 mM NaCl, 2 mM EDTA, and 0.5% [wt/vol] sodium deoxycholate) in water supplemented with protease and phosphatase inhibitors. Protein concentration was measured using bicinchoninic acid assay assay kit (Thermo Fisher Scientific) and normalised. The samples were then separated via SDS–PAGE and subsequently transferred onto a nitrocellulose membrane. Membranes were blocked in 3% (wt/vol) BSA or 5% (wt/vol) milk powder in TBS supplemented with 0.1% (vol/vol) Tween 20 and incubated with the primary and horseradish peroxidase secondary antibodies in blocking solution (see antibody list in Table S2).

Protein bands were visualised via chemiluminescence reaction using the Fusion-FX7 Advance imaging system (Peqlab Bio-technology). A densitometric analysis of at least three independent experiments was carried out using FIJI ImageJ 1.0 software.

### Cell viability assays

To assess cytotoxicity and morphological changes upon treatment with LecB, NHKs were seeded in a 24-well plate at a density of $5 \times 10^4$ per well and treated for 24 h with different LecB concentrations ± fucose (30 mM). Images were acquired using Evos FL Cell Imaging Systems (Thermo Fisher Scientific) using 20× objective (NA = 0.45).

MTT (3-(4,5-dimethylthiazol-2-yl)-2,5-diphenyltetrazolium bromide) tetrazolium reduction assay kit (Merck) was additionally used to quantify cell viability upon LecB treatment. For this purpose, cells were grown into a 96-well plate and stimulated for 24 h with the indicated LecB concentrations ± L-fucose. The MTT solution was added for 4 h at 37°C with 5% $CO_2$ and the absorbance at 570 nm was measured.

### Ultrastructural analyses

For conventional transmission electron microscopy, NHKs were treated or not with 5 $\mu$g/ml of LecB, for the indicated time periods. An equal volume of double-strength fixative (4% paraformaldehyde and 4% glutaraldehyde in 0.1 M sodium cacodylate buffer, pH 7.4) was then added to the cells for 20 min at room temperature before fixing the cells with one volume of single-strength fixative (2% paraformaldehyde and 2.5% glutaraldehyde in 0.1 M sodium cacodylate buffer, pH 7.4) for 2 h at room temperature. After washes with cacodylate buffer (pH 7.4), the cells were then scraped and embedded as previously described (Verheije et al, 2008). Ultrathin 70-nm sections were cut using a Leica EM UC7 ultra microtome (Leica Microsystems) and stained with uranyl acetate and lead citrate as previously described (Verheije et al, 2008).

For the IEM analyses, NHKs were treated for 12 h with biotinylated LecB and fixed by addition of 4% PFA and 0.4% glutaraldehyde in 0.1 M phosphate buffer (pH 7.4) in an equal volume to the DMEM medium, before to be incubated for 20 min at room temperature. The cells were subsequently fixed in 2% PFA and 0.2% glutaral-dehyde in 0.1 M phosphate buffer (pH 7.4) for 3 h at room temperature and then embedded for the Tokuyasu procedure before cutting into ultrathin cryosections and labelling them with immu-nogold, as previously described (Slot and Geuze, 2007). Biotinylated LecB was detected using an anti-biotin rabbit antibody (100-4198; Rockland). The labelling was revealed with protein A conjugated with 10 nm gold (CMC). As labelling control, NHKs were incubated with non-biotinylated LecB. No immunogold labelling was detected in this sample. The cell sections were analysed using a CM100 Bio TEM (FEI).

### Immunofluorescence of chronic human wounds

Fixed sections of human chronic wounds were kindly provided by D Kiritsi (Department of Dermatology, Medical Centre, Albert Ludwigs University, Freiburg). Antigens were retrieved with 0.05% pronase (Sigma-Aldrich). Sections were blocked with 3% (wt/vol) BSA in PBS and probed overnight with an anti-LecB or anti-*P. aeruginosa* antibody. After two washing steps with PBS supplemented with 0.1% Tween 20, the sections were stained with a dye-labelled secondary antibody, counterstained with DAPI, and mounted in fluorescence mounting medium (Dako). Images were acquired using Zeiss Axio Imager A1 fluorescence microscopy.

### Statistics

All data were obtained from at least three independent experi-ments and are shown as the means ± SEM.

Statistical analysis was performed using GraphPad Prism soft-ware. One-way or two-way ANOVA were chosen to assess signifi-cance in experiments with multiple conditions. Otherwise, when experimental data for one condition were compared with the relative control condition, multiple *t* test was applied.

A *P*-value < 0.05 was considered as statistically significant. * denotes *P* < 0.05; ** denotes *P* < 0.01; *** denotes *P* < 0.001; **** denotes *P* < 0.0001; ns denotes not significant.

## Supplementary Information

## Acknowledgements

The research group of W Römer was supported by the German Research Foundation (BIOSS—EXC 294, CIBSS—EXC-2189—Project ID 390939984, GSC-4, and RO 4341/2-1), the Ministry of Science, Research and the Arts Baden-Württemberg (Az: 33-7532.20), and by a starting grant from the European Research Council (Programme "Ideas", ERC-2011-Stg 282105-lec&lip2invade). The work in A Nyström's group was supported by the German Research Foundation (NY 90/5-1). F Reggiori is supported by ZonMW VICI (016.130.606), ZonMW TOP (91217002), ALW Open Programme (ALWOP.310), and Marie Skłodowska-Curie Cofund (713660) and Marie Skłodowska Curie ETN (765912) grants. M Mari is supported by an ALW Open Programme (ALWOP.355). This publication is partially based upon work from COST Action CA18103 (INNOGLY), supported by COST (European Cooperation in Science and Technology). The article processing charge was funded by the German Research Foundation (DFG) and the University of Freiburg in the funding programme Open Access Publishing. We appreciate the support by the BIOSS Toolbox, especially P Salavei, in the purification of the bacterial lectins.

## Author Contributions

A Landi: conceptualization, data curation, investigation, methodology, and writing—original draft, review, and editing.
M Mari: conceptualization, data curation, investigation, and writing—review and editing.
S Kleiser: data curation and investigation.
T Wolf: investigation and methodology.
C Gretzmeier: investigation.
I Wilhelm: investigation.
D Kiritsi: methodology and writing—review and editing.
R Thünauer: investigation.
R Geiger: supervision and methodology.
A Nyström: conceptualization, supervision, and writing—review and editing.
F Reggiori: conceptualization, supervision, methodology, and writing—review and editing.
J Claudinon: investigation.
W Römer: conceptualization, supervision, funding acquisition, and writing—review and editing.

## Conflict of Interest Statement

The authors declare that they have no conflict of interest.

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
