## [Reviewer comments · Life Science Alliance]

Life Science Alliance

P. aeruginosa lectin LecB impairs keratinocyte fitness by abrogating growth factor signalling

Alessia Landi, Muriel Mari, Svenja Kleiser, Tobias Wolf, Christine Gretzmeier, Isabel Wilhelm, Dimitra Kiritsi, Roland Thünauer, Roger Geiger, Alexander Nyström, Fulvio Reggiori, Julie Claudinon, and Winfried Römer

DOI: <https://doi.org/10.26508/lsa.201900422>

Corresponding author(s): Winfried Römer, Albert-Ludwigs-University Freiburg

Review Timeline:

Submission Date:	2019-05-10
Editorial Decision:	2019-06-11
Revision Received:	2019-09-27
Editorial Decision:	2019-10-09
Revision Received:	2019-10-15
Accepted:	2019-10-16

Scientific Editor: Andrea Leibfried

Transaction Report:

June 11, 2019

Re: Life Science Alliance manuscript #LSA-2019-00422-T

Prof. Winfried Römer
Albert-Ludwigs-University Freiburg
Faculty of Biology, BIOS Centre for Biological Signalling Studies
Schänzlestraße 1
Schänzlestraße 18
Freiburg 79104
Germany

Dear Dr. Römer,

Thank you for submitting your manuscript entitled "Pseudomonas aeruginosa lectin LecB impairs keratinocyte fitness by abrogating growth factor signalling" to Life Science Alliance. The manuscript was assessed by expert reviewers, whose comments are appended to this letter.

As you will see, the reviewers appreciate your analyses and provide constructive input on how to further strengthen your work. We would thus like to invite you to submit a revised version of your manuscript, addressing the individual comments made by the reviewers. The revision requests aim at providing better support for your main conclusions and seem all straightforward to address, but please get in touch in case you would like to discuss individual points further with us.

Thank you for this interesting contribution to Life Science Alliance. We are looking forward to receiving your revised manuscript.

Sincerely,

B. MANUSCRIPT ORGANIZATION AND FORMATTING:

Reviewer #1 (Comments to the Authors (Required)):

Landi et al. studied the effects of the exposure of Lectin B (LecB) from *P. aeruginosa* on human keratinocytes. LecB associates with IGF-1R (and possibly with other RTKs, like EGFR) and downregulates its surface level upon long term treatment, without causing receptor kinase activation. It also causes a decreased Erk signaling activation and cell cycle arrest. The authors

proposed that LecB induces internalization of IGF-1R and its trafficking towards a lamp1-positive, LC3-positive, compartment, leading to reduced signaling.

The manuscript tackled an important issue in cell biology, and unveils an interesting and novel mechanism of action of LecB to dampen proliferation and survival in the host cells. However, there are a number of issues listed below that will need to be solved prior publication.

- 1) A quantitative assessment of IGF-1R surface levels upon treatment with Lec B is needed as the PM staining in Fig. 2B are not totally clear, with a lot of background signal. A more quantitative measurement of IGF-1R surface levels at steady state and upon Lec B treatment at different time points should be provided (e.g. by FACS). A parallel treatment with L-fucose would be a critical control for LecB specificity in causing IGF-1R downmodulation from the cell surface.
- 2) A time course of intracellular IGF-1R accumulation upon LecB treatment should also be shown in addition to surface to prove that disappearance from the PM is paralleled by the IGF-1R accumulation in intracellular compartments (i.e. endocytosis).
- 3) The authors suggest that IGF-1R might be degraded upon LecB, as they showed that the IGF-1R fraction bound to LecB decreased over-time (Fig. S6A). This is a very interesting and important data that should be moved in main figures. However, authors need to prove that this decreased is really due to degradation by performing treatment with lysosome inhibitor (e.g. chloroquine or Bafilomycin) to show that IGF-1R levels are restored. Did the authors check whether total IGF-1R levels (not only the bound fraction) decrease over time upon LecB incubation? Please show total levels of IGF-1R receptor in Fig. 3B.
- 4) The data that the fraction of ubiquitinated protein bound to LecB increased over time is intriguing (Fig. S6B). Is IGF-1R itself ubiquitinated upon LecB? This would be a very interesting data as it would indicate that the mechanism of LecB induced endocytosis of IGF1R could be mediated by ubiquitin, which is indeed a critical internalization and sorting signal toward a degradative compartment.
- 5) The data on LC3 are very interesting, but incomplete. The authors proposed that LecB mediates LC3-associated phagocytosis (LAP), as they observed an increase in LC3-II but they could not observe autophagosomal double-membrane structures. To prove this, authors should show, for instance, that their pathway is sensitive to Vps34-KD (or to phagocytosis inhibitors) but not to inhibition of autophagosome formation (treatment with 3-Methyladenine or FIP200 KD, see Florey et al, NCB 2011).

Minor points

- 1) Please note that the acronym for Epidermal Growth Factor Receptor is EGFR and not EGF-1R.
- 2) Since the other RTKs found in the mass spec are not further investigated in the manuscript, I would change the sentence in the abstract and along the manuscript 'LecB associates with several growth factor receptors and dampens their signaling pathway', as it is not shown.
- 3) In Fig. 1A, please explain why there's PAO1 signal in normal skin. Is also this sample infected by *P. aeruginosa*?

Reviewer #2 (Comments to the Authors (Required)):

Reviewing MS Ref: LSA-2019-00422

Title: *Pseudomonas aeruginosa* lectin LecB impairs keratinocyte fitness by abrogating growth factor signalling

Journal: Life Science Alliance

Authors: Alessia Landi, Muriel Mari, Tobias Wolf, Svenja Kleiser, Christine Gretzmeier, Isabel Wilhelm,

Dimitra Kiritsi, Roland Thünauer, Roger Geiger, Alexander Nytröm, Fulvio Reggiori, Julie Claudinon, and Winfried Römer

This work focuses on the effect of *P. aeruginosa* fucose-binding lectin LecB on human keratinocytes. They found that LecB co-immunoprecipitated with several growth factor receptors : EGFR, Met, and IGF1,2R. They also showed that LecB induced IGF-1R internalization without activating the receptor (MAPK, mTor), in contrast to its natural ligand (IGF-1). Instead LecB induced cell cycle arrest. Finally the authors investigated LecB-mediated endocytosis and showed that it induced LC3 positive vesicles where IGF-1R co-localized and that LecB could also be found associated to Lamp1 and Rab9 but not to recycling endosomes. Consistently, EM data showed that LecB was associated to membrane ruffles and internalized into degradative vesicles especially in intraluminal vesicles. Overall the study proposes a new function of LecB that would be to induce the endocytosis and degradation of some signaling receptors, hence switching off keratinocyte proliferation and survival.

The work is very interesting, although it does not describe the full mechanism of LecB action. Moreover, each point sum up above are supported by strong data.

I have several comments about the works:

1-Fig. 1A: I don't understand why in normal skin there is a PAO1 (*P. aeruginosa*) staining, is that a strong PAO1 background?

The authors either should explain this labeling.

2-TableS1: I am not sure to fully understand the negative control of this experiment. To what I understood, the table S1 lists proteins that were seen differently co-IPed with LecB between treated and untreated cells. If this is the case, I would like to see the data for proteins not enriched upon treatment with LecB. Is there other membrane receptors associated to LecB even those interacting to a lesser extend?

This should be better described and provide some control of proteins not enriched upon LecB treatment.

3- About the endocytosis part, I wonder whether this LecB-induced endocytosis and sorting of the receptors to the degradation can be used by other receptors than IGF1R, Met, EGFR. Indeed, these 3 signaling receptors are known to be endocytosed via a clathrin-independent process (Boucrot et al, 2015). It would be very interesting to determine if other cargo like Transferrin receptor (TfR, endocytosed by a clathrin-dependent route, not signaling and always recycling) is uptaken into LC3-positive vesicles upon LecB treatment. This will allow at drawing a clearer mechanism of LecB action on signalling receptor trafficking pathway.

In Fig. 5 the authors could add another experiment showing TfR endocytosis and intracellular trafficking upon LecB treatment as compared to IGF1R.

Reviewer #1

Landi et al. studied the effects of the exposure of Lectin B (LecB) from P. aeruginosa on human keratinocytes. LecB associates with IGF-1R (and possibly with other RTKs, like EGFR) and downregulates its surface level upon long term treatment, without causing receptor kinase activation. It also causes a decreased Erk signalling activation and cell cycle arrest. The authors proposed that LecB induces internalization of IGF-1R and its trafficking towards a lamp1-positive, LC3-positive, compartment, leading to reduced signaling.

The manuscript tackled an important issue in cell biology, and unveils an interesting and novel mechanism of action of LecB to dampen proliferation and survival in the host cells. However, there are a number of issues listed below that will need to be solved prior publication.

- 1) A quantitative assessment of IGF-1R surface levels upon treatment with LecB is needed as the PM staining in Fig. 2B are not totally clear, with a lot of background signal. A more quantitative measurement of IGF-1R surface levels at steady state and upon LecB treatment at different time points should be provided (e.g. by FACS). A parallel treatment with L-fucose would be a critical control for LecB specificity in causing IGF-1R downmodulation from the cell surface.*

We appreciate the experimental suggestion. However, it is not possible to investigate the surface expression of IGF-1R via FACS, as LecB induces a strong attachment of the cells to plates (even after 30 minutes exposure), making cell detachment only possible by scraping, which affects the integrity of the plasma membrane. Therefore, we employed cell surface biotinylation to measure the surface and intracellular levels of the receptor at any given time. These experiments confirmed a time-dependent loss of the receptor from the plasma membrane, which can be blocked by L-fucose supplementation (Fig S2A, B).

- 2) A time course of intracellular IGF-1R accumulation upon LecB treatment should also be shown in addition to surface to prove that disappearance from the PM is*

paralleled by the IGF-1R accumulation in intracellular compartments (i.e. endocytosis).

Combining cell surface biotinylation and western blot, we had respectively quantified the intracellular receptor levels upon LecB treatment and discriminated between IGF-1R and its precursor form. These results show an intracellular decrease of IGF-1R, which is concomitant with an increase of the precursor form. This suggests, on one hand, a degradation of the receptor and on the other, the activation of a cellular response to compensate the receptor depletion (Fig S2C, D).

Besides, immunofluorescence experiments showed both a redistribution of the receptor, which forms accumulations in LAMP-1-positive compartments over time, and a strong increase of lysosomes upon LecB treatment (Fig S1C, D).

- 1) *The authors suggest that IGF-1R might be degraded upon LecB, as they showed that the IGF-1R fraction bound to LecB decreased over-time (Fig. S6A). This is a very interesting and important data that should be moved in main figures. However, authors need to prove that this decreased is really due to degradation by performing treatment with lysosome inhibitor (e.g. chloroquine or Bafilomycin) to show that IGF-1R levels are restored. Did the authors check whether total IGF-1R levels (not only the bound fraction) decrease over time upon LecB incubation? Please show total levels of IGF-1R receptor in Fig. 3B.*

As suggested, we moved the Fig S6A, B into main figures (Fig 5C, D) and we added the pan levels related to the phospho-blot in Fig 2D.

To address whether the decrease of IGF-1R is due to lysosomal degradation, we blocked lysosomal turnover with bafilomycin A1. As expected, treatment with bafilomycin A1 alone increased the IGF-1R levels. Combined treatment of bafilomycin A1 and LecB after 3 h partially counteracted LecB-evoked IGF-1R degradation. These new results demonstrate a decrease of the total levels of the receptor after 3 h of LecB treatment, which is only partially restored by bafilomycin A1 treatment (Fig S1E, F). The incomplete recovery indicates that LecB might use additional pathways, to lysosomal degradation (e.g. proteasome), to deplete the receptor.

- 2) *The data that the fraction of ubiquitinated protein bound to LecB increased over time is intriguing (Fig. S6B). Is IGF-1R itself ubiquitinated upon LecB? This would be a very interesting data as it would indicate that the mechanism of LecB induced endocytosis of IGF1R could be mediated by ubiquitin, which is indeed a critical internalization and sorting signal toward a degradative compartment.*

By immunoprecipitating IGF-1R, we could indeed detect ubiquitin in the eluted fraction 1 h after exposure with LecB (Fig S6C, D), together with a decrease of IGF-1R (and an increase of the precursor). This indicates that ubiquitin might be one sorting signal for receptor internalisation and degradation.

- 3) *The data on LC3 are very interesting, but incomplete. The authors proposed that LecB mediates LC3-associated phagocytosis (LAP), as they observed an increase in LC3-II but they could not observe autophagosomal double-membrane structures. To prove this, authors should show, for instance, that their pathway is sensitive to Vps34-KD (or to phagocytosis inhibitors) but not to inhibition of autophagosome*

formation (treatment with 3-Methyladenine or FIP200 KD, see Florey et al, NCB 2011).

Like FIP200, ATG13 is a component of the ULK kinase complex, which is essential for autophagosome formation but dispensable for LAP. Therefore, we silenced ATG13 using siRNA before treating cells with LecB (Fig S7 E-G). This new experiment revealed that stimulation of LC3-II formation upon cell exposure to LecB is ATG13 independent, further excluding the involvement of autophagy. Moreover, to elucidate whether this pathway is sensitive to phagocytosis inhibitors, we treated the cells with cytochalasin D, a compound that blocks actin polymerisation and thus phagocytosis. As we observed a reduction of LecB uptake in the cytochalasin D-treated cell, we concluded that the LecB uptake process is actin-dependent (Fig S8 A, B).

Minor points

1) Please note that the acronym for Epidermal Growth Factor Receptor is EGFR and not EGF-1R.

We apologise for the mistake and correct the acronym.

2) Since the other RTKs found in the mass spec are not further investigated in the manuscript, I would change the sentence in the abstract and along the manuscript 'LecB associates with several growth factor receptors and dampens their signalling pathway', as it is not shown.

We agree with the reviewer and change the sentence into “LecB associates with insulin like growth factor receptor 1, dampening its signalling pathways”.

3) In Fig. 1A, please explain why there's PAO1 signal in normal skin. Is also this sample infected by P. aeruginosa?

The normal skin is not infected with PAO1. The signal appearing is a strong background of the PAO1 antibody. For clarification, we show here below a zoomed picture with the separate channels of the two stainings (infected wound section versus normal skin) and added an explanation in Fig 1 legend. The green signal, indicated by an arrow in the normal skin section, is due to the *stratum corneum*, which often is responsible for unspecific staining with some antibodies. The chronic wounds did not have a well-developed *stratum corneum*, leading to the absence of this unspecific signal in these samples.

Reviewer #2

This work focuses on the effect of P. aeruginosa fucose-binding lectin LecB on human keratinocytes. They found that LecB co-immunoprecipitated with several growth factor receptors: EGFR, Met, and IGF1,2R. They also showed that LecB induced IGF-1R internalization without activating the receptor (MAPK, mTor), in contrast to its natural ligand (IGF-1). Instead LecB induced cell cycle arrest. Finally the authors investigated LecB-mediated endocytosis and showed that it induced LC3 positive vesicles where IGF-1R co-localized and that LecB could also be found associated to Lamp1 and Rab9 but not to recycling endosomes. Consistently, EM data showed that LecB was associated to membrane ruffles and internalized into degradative vesicles especially in intraluminal vesicles. Overall the study proposes a new function of LecB that would be to induce the endocytosis and degradation of some signalling receptors, hence switching off keratinocyte proliferation and survival.

The work is very interesting, although it does not describe the full mechanism of LecB action. Moreover, each point sum up above are supported by strong data.

I have several comments about the works:

- 1) *Fig. 1A: I don't understand why in normal skin there is a PAO1 (P. aeruginosa) staining, is that a strong PAO1 background? The authors either should explain this labeling.*

It is a strong background signal in the stratum corneum, as specified in the minor point 3 of the reviewer #1.

- 2) *TableS1: I am not sure to fully understand the negative control of this experiment. To what I understood, the table S1 lists proteins that were seen differently co-IPed with LecB between treated and untreated cells. If this is the case, I would like to see the*

data for proteins not enriched upon treatment with LecB. Is there other membrane receptors associated to LecB even those interacting to a lesser extend? This should be better described and provide some control of proteins not enriched upon LecB treatment.

We apologise for the lack of clarity. Likely, due to the high abundance of fucosylated receptors, other surface proteins are targeted by LecB. Here, we focus on IGF-1R due to its high fold enrichment in the pull-down fractions. However, EGFR was also found to be co-immunoprecipitated with LecB (Fig S1A, B), but it was less enriched in the LecB treated cells, in comparison to IGF-1R (difference untreated-treated EGFR = -3.81 versus IGF-1R = -7.95). We have added GAPDH as a negative control for the pull-down assay (Fig S1B). As an additional, we checked whether transferrin receptor (TfR) was co-isolated with LecB included because its trafficking was found unaffected upon LecB treatment (please see below the related answer). Interestingly, we did not find it in the LecB treated fraction, indicating that is not an interacting partner (Fig S9B).

3) About the endocytosis part, I wonder whether this LecB-induced endocytosis and sorting of the receptors to the degradation can be used by other receptors than IGF1R, Met, EGFR. Indeed, these 3 signaling receptors are known to be endocytosed via a clathrin-independent process (Boucrot et al, 2015). It would be very interesting to determine if other cargo like Transferrin receptor (TfR, endocytosed by a clathrin-dependent route, not signaling and always recycling) is uptaken into LC3-positive vesicles upon LecB treatment. This will allow at drawing a clearer mechanism of LecB action on signalling receptor trafficking pathway. In Fig. 5 the authors could add another experiment showing TfR endocytosis and intracellular trafficking upon LecB treatment as compared to IGF1R.

We thank the reviewer for the suggestion. Indeed, we performed immunofluorescence experiments to investigate TfR trafficking upon cell treatment with LecB to monitor whether it colocalises with LC3 positive vesicles. Interestingly, we could not observe colocalisation between LC3 and TfR (Fig 6D, E), which indicates that LecB does not target the entire recycling route, but it rather induces receptor depletion to abrogate signalling. These data are supported by biochemical analysis of co-precipitation (Fig S9B)

October 9, 2019

RE: Life Science Alliance Manuscript #LSA-2019-00422-TR

Prof. Winfried Römer
Albert-Ludwigs-University Freiburg
Faculty of Biology, BIOSS Centre for Biological Signalling Studies
Schänzlestraße 1
Schänzlestraße 18
Freiburg 79104
Germany

Dear Dr. Römer,

Thank you for submitting your revised manuscript entitled "P. aeruginosa lectin LecB impairs keratinocyte fitness by abrogating growth factor signalling". As you will see, the reviewers appreciate the introduced changes, and we would thus be happy to publish your paper in Life Science Alliance pending final minor revisions mainly necessary to meet our formatting guidelines:

- Please address the remaining concerns of rev#1
- Please adjust the author order in our submission system so that it matches the one in your ms file
- Please incorporate the supplementary procedures in the main ms text
- Please deposit the mass-spec data to a repository to allow re-analysis by others
- Please make sure that all figure panels are correctly called-out (change callout to S5A-E to S5A-E', remove callout to 4B', add callout to 6D-E)

A. FINAL FILES:

B. MANUSCRIPT ORGANIZATION AND FORMATTING:

Sincerely,

Reviewer #1 (Comments to the Authors (Required)):

In the revised version of the manuscript, authors performed a series of additional experiments and were able to reply to almost all the concerns that I raised in the first round of revision. There are only two remaining issues that are not convincing to me, listed below, and I suggest to remove the correspondent experiment and/or to discuss it further. For the rest, the manuscript is now suitable for publication in LSA.

- 1) The Ubiquitination experiment in Fig. S6D is not convincing, I'm not even sure about specificity of the signal. I would simply remove it.
- 2) The biotinylation-based internalization assay in Fig. S2 seems strange to me: on the basis of the immunofluorescence experiments, I would expect to see a decrease in surface IGF-1R upon increasing time of treatment with Lectin B (as it is observed), paralleled by an increase in intracellular (=internalized) IGF-1R, at least from zero to 30 min time point. On the contrary, a decreased is observed. Can the authors explain this better? Why there is more intracellular IGF-1R at time zero (=no internalization) than at 30 min? Is Lectin B also stimulating degradation of IGF-1R in the biosynthetic pathway? If the result is unclear I would suggest to remove it. There are now enough experiments in the same direction.

Reviewer #2 (Comments to the Authors (Required)):

The revised version of manuscript written by Laten et al (MS Ref: LSA-2019-00422) entitled: *Pseudomonas aeruginosa* lectin LecB impairs keratinocyte fitness by abrogating growth factor signalling, improved a lot from the initial version and the authors answered to all referees' issues. Therefore, the manuscript should be published.

- 1) We have removed the figures S2 and S6C-D, as suggested by reviewer #1, and adjusted the figure's numbers accordingly.
- 2) The author order was adjusted in the system.
- 3) The mass-spec data were uploaded in the system.
- 4) We have corrected the call-out 4B' and added the call-out 6D-E.

October 16, 2019

RE: Life Science Alliance Manuscript #LSA-2019-00422-TRR

Prof. Winfried Römer
Albert-Ludwigs-University Freiburg
Faculty of Biology, BIOS Centre for Biological Signalling Studies
Schänzlestraße 18
Freiburg 79104
Germany

Dear Dr. Römer,

Thank you for submitting your Research Article entitled "P. aeruginosa lectin LecB impairs keratinocyte fitness by abrogating growth factor signalling". It is a pleasure to let you know that your manuscript is now accepted for publication in Life Science Alliance. Congratulations on this interesting work.

DISTRIBUTION OF MATERIALS:

Again, congratulations on a very nice paper. I hope you found the review process to be constructive and are pleased with how the manuscript was handled editorially. We look forward to future exciting submissions from your lab.

Sincerely,

Andrea Leibfried, PhD
Executive Editor
Life Science Alliance
Meyerohofstr. 1
69117 Heidelberg, Germany
t +49 6221 8891 502
e a.leibfried@life-science-alliance.org
www.life-science-alliance.org